# Analysis of the Fire Behavior of Polymers (PP, PA 6 and PE-LD) and Their Improvement Using Various Flame Retardants

**DOI:** 10.3390/ma13245756

**Published:** 2020-12-16

**Authors:** Dieter Hohenwarter, Hannelore Mattausch, Christopher Fischer, Matthias Berger, Bernd Haar

**Affiliations:** 1Federal Testing Center TGM, Department of Plastics Technology and Environmental Engineering, Wexstrasse 19-23, 1200 Wien, Austria; mberger@tgm.ac.at; 2Montanuniversitaet Leoben, Polymer Processing, Otto-Gloeckel-Strasse 2, 8700 Leoben, Austria; hannelore.mattausch@gmail.com; 3Laboratory for Polymer Engineering (LKT) at TGM, Wexstrasse 19-23, 1200 Wien, Austria; cfischer@tgm.ac.at; 4Polymer Competence Center Leoben GmbH, Roseggerstraße 12, 8700 Leoben, Austria; bernd.haar@pccl.at

**Keywords:** cone calorimeter, UL 94, polyethylene, polypropylene, polyamide, flame retardants, crosslinking, aging

## Abstract

The fire behavior of polymers is examined primarily with the time-dependent heat release rate (HRR) measured with a cone calorimeter. The HRR is used to examine the fire behavior of materials with and without flame retardants, especially Polypropylene (PP-Copo) and Polyethylene (PE-LD). Polypropylene is stored for up to 99 days under normal conditions and the heat release rate shows especially changes about 100 s after irradiation with cone calorimeter, which may be caused by aging effects. The effect of crosslinking to the burning behavior of PP was examined too. Polyamides (PA 6) are irradiated with a radiation intensity of 25 kW/m^2^ to 95 kW/m^2^ and fire-related principles between radiation intensity and time to ignition can be derived from the measurement results. In order to comprehensively investigate the fire behavior of PP (also with flame retardant additives), the samples were also exposed to a flame, according to UL 94 with small power (50 W) and is inflamed with the power of a few 100 W. The irradiation causes different trigger mechanisms for the flame retardant additives in a plastic than the flame exposure. It is shown that the compound, which is favorable for irradiation, is not necessarily good for flame exposure. It can be seen that expandable graphite alone or with the addition of other additives is a very effective flame retardant for PP.

## 1. Introduction

This paper presents the results of investigations into the fire behavior of polymeric materials. These measurements were mainly made with the cone calorimeter according to ISO 5660-1 [1], in which the material can be irradiated with an intensity in the range of 25 kW/m^2^ to 100 kW/m^2^.

It was the aim to investigate the influence of different parameters, e.g., the aging of polypropylene, the influence of the radiation intensity and of various flame retardants. Furthermore, PP (copolymer), PA 6 (extrusion grade) and various PE-LD types (unmodified and modified) were tested to give an overview about their different burning behavior. It has to be stated that not all variations were determined for each polymer as this would exceed the scope of this paper.

The given data were obtained during various academic projects at the Montanuniversitaet Leoben (Technical University of Leoben, Austria) and the Laboratory for Polymer Engineering (LKT, Vienna, Austria) in cooperation with the plastics technology department of the technical college TGM at Vienna. The focus of these projects was to determine the basic fire behavior of polymers.

The results presented in this paper show that the heat release rate can be used to examine very sensitive material changes (aging of polypropylene [2,3], for example) as a function of time. The time dependent heat release rate (HRR) is used to describe the burning behavior of the material by means of diagrams. The analysis of the measurement results (heat release rate HRR) enables conclusions to be drawn about material properties depending on time. When examining material properties with the cone calorimeter, the effect of flame retardants on polymers can also be investigated. It should be noted, however, that, in this case, the material is exposed to thermal radiation. This can lead to different results than exposition to a direct flame.

Hence, for an extensive assessment of fire-related material properties, it may also be required to stress the material directly with a flame. In order to comprehensively assess various flame retardants for polymers, regarding the exposition to thermal radiation (cone calorimeter test) or exposure to flame, flame tests have to be conducted too.

Tests on polymers with different flame retardants result in a sequence of the effectiveness of the flame retardants when irradiated with the cone calorimeter. A different order of effectiveness may result in flame exposure tests (UL 94). The reasons for this are the different trigger mechanisms of irradiation or flame exposure.

The book “Plastics Flammability Handbook” by Jürgen Troitzsch [4] offers a very good overview of the flammability of polymers and also of the various fire characterization methods in Europe and the US.

A work on the fire behavior of various materials with regard to their suitability for fire-related use in rail vehicles is included in reference [5]. A work that focuses on the production of transparent components made of polycarbonate via 3D-printing is included in reference [6]. Processing and printing of the polymer causes thermal stress to the molecules. This may lead to a worsening of the flammability causing a decline of the properties compared to the virgin polycarbonate. In summary, the printing parameters and additives must be carefully considered in relation to the fire properties of the polymer [6]. Furthermore, an overview about the effect of important flame retardants, as used as additives, is given in the references. The chapter “Flame Retardants” in the “Plastics Additives Handbook” by Zweifel, et al. [7] gives a useful description of the mode of action of various flame retardants. The effect of aluminum hydroxide, magnesium hydroxide and aluminium phosphinate is described in [8,9,10,11]. Additionally, Refs. [12,13,14,15] give detailed information about the effect of inorganic minerals like montmorillonite, kaolinite, zeolite and clay. Ref. [16] describes the effect of aluminium diethylphosphinate and melamine polyphosphate in conjunction with organically modified montmorillonite nanoclay on PA 6 in detail. Deeper insight about the effect of radiation crosslinking on flame retarded polymers is given in [17], while the effect of graphene, expanded and expandable graphite on polypropylene, polyamide 6 and polylactid acid is described in [18,19,20,21].

The named flame retardants are commonly used for the improvement of the burning behaviour of polymers. Hence, those additives were chosen for the investigation of their specific influence on the heat release rate, especially of polypropylene. While inorganic compounds as aluminum hydroxide or magnesium hydroxide have to be used in higher percentages, expandable graphite is known to improve the burning behavior at low percentages. Additionally, inorganic additives as montmorillonite or zeolite cause higher density and abrasion during processing. To investigate possible alternatives, the former mentioned additives were used and compared with commonly used flame retardants like melamine polyphosphate, for example.

## 2. Materials and Methods

The materials used are specified below (see Table 1). The used polypropylene grade BB412E (copolymer) (Borealis, Vienna, Austria) is recommended for pipe extrusion and injection molding of fittings in the field of waste water discharge and sewage.

### 2.1. Sample Preparation

The granulate was produced to plates with a vacuum laboratory press P200PV (Collin GmbH, Maitenbeth, Germany) at about 170 °C for about 60 min (pressing pressure about 150 bar). In order to investigate if the manufacturing process has an effect on the heat release, the plates were manufactured by pressing and injection molding. Injection molding was performed with an electric injection molding machine (EM1600/350, Battenfeld, Kottingbrunn, Austria), with a plate tool. The injection pressure was on the order of 700 bar and a holding pressure of 250 bar was applied for 5 s. The specimen geometry was comparable to the pressed plates, with dimensions of 120 mm × 120 mm × 2 mm. Processing temperature for PP was in the range of 220 °C and mold temperature was 40 °C.

### 2.2. Determination of the Fire Behavior

The heat release measurements (HRR) were made with the cone calorimeter according to ISO 5660-1 [1], in which the material can be irradiated with an intensity of 25 kW/m^2^ to 100 kW/m^2^ and the resulting gases are ignited with an electrical ignition spark. The used cone calorimeter was produced by Fire Testing Technology (FTT) (West Sussex, UK) in the year 2007. According to the ISO 5660-1 standard, 3 specimens of the sample must be measured, and the mean value determined from the results. Due to the large number of measurements and the sometimes-complicated production of test samples, for some samples, just one or two measurements were conducted.

The cone calorimeter test method is based on the observation that the net heat of combustion is proportional to the amount of oxygen required for combustion. The rate of heat release is determined by the fact that 13,100 J of heat are released per kilogram of oxygen consumed. The oxygen content of the exhaust gases is measured during the entire investigation and the amount of heat released is calculated as a function of time with a time interval of 2 s.

The fire behavior of the material is determined with radiation and with flaming according to UL 94 [22]. In the case of flame exposure according to the UL 94 test, flame exposure of the vertically arranged test sample (UL 94 V) is common, but the result is only a passed or failed classification. Therefore, the UL 94 HB (Horizontal Burning) test with horizontal test specimen arrangement was chosen [22], where the burning rate is indicated as the result and thus a finer distinction between the individual material tests is possible (see Figure 1).

The UL 94 tests were carried out with an ATLAS testing device (HVUL2, Atlas Material Testing Technology, Chicago, IL, USA) corresponding to the European standard EN 60 695-11-10 [23]. The material is exposed to a burner power of 50 W. A burner tube with an inner diameter of 9.5 mm is used and then 50 W corresponds to a flame length of 20 mm. The test rod has a length of 125 mm and a width of 13 mm and is exposed to a flame and the burning rate is determined in the range between 25 mm and 100 mm (see [23]). The thickness of the test samples is different and was 3 mm, unless otherwise stated. The UL 94 test is very common in electrotechnical and automotive engineering.

In Austria, the flame resistance (called Schwerbrennbarkeit) was investigated in earlier years using the Schlyter test (ÖNORM A 3800 [24]), where two parallel plates (length 80 cm, width 30 cm) are 5 cm apart and one of them is exposed to a flame. The burner of the Schlyter test is 17 cm wide and has a total burner output of 1700 Watt, i.e., 100 Watt/cm. In Germany, this test is called not easily flammable (Schwerentflammbar, DIN 4102-1 [25]), and there are 4 plates arranged in a square and loaded by the flame. Although the flame tests are different, the classifications are identical for many materials in Austria and Germany.

Today, the Schlyter test has been replaced by EN tests, and the product is classified according to ÖNORM EN 13501-1 [26]. For these tests, however, much larger test samples are required, which are set up in a corner.

### 2.3. Precision of Fire Behavior Tests

For the comparison of cone tests with UL 94 tests, it is shown in reference [27] that, especially at low irradiance levels (30 kW), a kind of correlation between cone and UL 94 V results could be found. In Ref. [27], it is claimed for 2006 that the UL 94 V tests are prone to unpredictable results. The results of interlaboratory tests with the Cone Calorimeter [28] and the results of UL 94 tests [29] show that, with 10 participating laboratories, the results of the mean values of 9 test sites agree quite well. Thus, it is today that the results of fire tests are well secured in international comparison.

### 2.4. Differential Scanning Calorimetry

The example of PP is also used to investigate whether the degree of crystallization influences the fire behavior. This change in crystallinity was verified by Differential Scanning Calorimetry (DSC), according to ÖNORM EN ISO 11357-3 [30]. The crystallinity was calculated by determination of the melting enthalpy in the first heating run, using a heating rate of 20 K/min. The melting enthalpy was referred to a melt enthalpy of a one hundred percent crystallin PP-Homo with 207 J/g [31].

## 3. Results and Discussion

### 3.1. Burning Behavior of Polymeric Materials without Flame Retardant

In this section, the fire-related results of the investigation of polypropylene (PP) and polyamide (PA) are presented. The changes in burning behavior of polypropylene (PP) were detected by cone calorimeter, from the day after production to day 99 stored at room temperature. Furthermore, it is shown that different sample weights cause different heat release rates, and different irradiation intensities lead to different ignition times. This physical/chemical relationship is illustrated in Section 3.1.2 using the example of polyamide. It is also demonstrated that low density polyethylene (PE-LD) from different manufacturers cause slightly different heat release rates.

#### 3.1.1. Aging of Polypropylene (PP)

For the following tests, PP granulate was pressed with a laboratory press and examined on different days (one day to up to 99 days) after production. The mass of the pressed plates was 50 g. The investigations were carried out with the cone calorimeter and the heat release rate was determined as a function of time.

However, only plates with a thickness of 2 mm could be processed by injection molding, which is the reason why the specimen mass deviates from the pressed samples, with a mass of only 20 g. Hence, lower heat release rates (HRR) were achieved.

Figure 2 shows the slightly different heat release of three test samples on the first day after production. In this case, the test samples were made of pressed granules (m = 50 g).

Figure 3 displays the different heat release rates of PP plates produced by injection molding, which were measured on the day of production and 33 days later. The test samples were stored at room temperature and the mean values of three measurements are shown. Above all, the different heat release rates around the 70th second indicate a slight material change in the test sample, which is probably caused by aging. However, the results obtained (MARHE, HRRpeak) are not influenced significantly. The change of the characteristic of the curve progression is clearly visible due to a local maximum of the red curve (indicating the samples after 33 days) at about 70 s after the start of measurement.

Figure 4 displays the different heat release rates, which were measured on the day after production and finally after day 99. The analyses result on the different days are the mean of three measurements. The diagram shows that the material changes in terms of burning behavior, which is visible in the initial phase (indicated by red circle in Figure 4) through the change in the peak around 100 s after the start of the measurement. As the fire progresses, the maximum heat release rate changes as well, which is represented by HRRpeak. One possible change in polypropylene is the aging of the material, which can undergo post-crystallization through storage [2,3].

To check if the degree of crystallization influences the fire behavior of PP, injection molded specimens were stored at 80 °C for five days. It was supposed that the crystallinity would rise due to the tempering. This change in crystallinity was verified by Differential Scanning Calorimetry (DSC). Samples that were tested shortly after production and after five days. To indicate the change in crystallinity depending on the storing temperature, samples were thermally conditioned at 8 °C, at room temperature (23 °C and 50% relative humidity, according to ISO 291 [32]) and at 80 °C. The obtained values for the crystallinity of PP are given in Table 2.

The given values represent the mean value of three measurements, which were obtained across the sample thickness of 4 mm (see Figure 5).

It can be seen in Table 2 that none of the storing conditions resulted in a significant change of crystallinity. Therefore, differences in the burning behavior of PP will not be caused by post-crystallization.

The tempered samples were compared to non-tempered samples using the Cone-Calorimeter, since PP is well known to be sensitive for aging. The results of this comparison display only minor differences in heat-stored and non-heat-stored specimens. The slight difference between the two series indicates that there was actually no post-crystallization and no additives (e.g., product stabilizers) that had migrated. Since there was no post-crystallization verifiable, these results correlate with the DSC-measurements shown in Table 2. Changes of the curve characteristic may be caused by a loss of heat stabilizers or other aging effects that could not be determined.

As it was seen in Figure 4, the peak at around 100 s (after the start of irradiation) presents differences in heat release after one day and 99 days. In Figure 6, the measurement results of day 1 (as mean values) and some single results of day 99 and the mean value, shown as a grey curve, are compared.

So far, the presented results show that material changes of PP can be detected very sensitively with the cone calorimeter by heat release rate measurements.

#### 3.1.2. Effect of Mass on the Heat Release of Polyamide (PA 6)

In this section, different masses of polyamide (PA 6) specimens were tested with a radiation intensity of 50 kW/m^2^. It was investigated how the weight differences affect the fire behavior of PA 6 (see Figure 7).

Figure 7 describes the time dependency of the heat release rate. The maximum value of the HRR is included in the HRR peak. The maximum average rate of heat emission (MARHE) is a single number used to describe the maximum heat release and is also included as a limit value in regulations, e.g., for railways in EN 45 545-2. The effective heat of combustion (EHC) and the time to ignition are fire engineering material characteristics shown in Table 3.

It can be seen that the heat release from the same material (polyamide) has a different curve shape. It is noticeable that, with lower sample mass, two peaks occur, while the larger mass burns very continuously. The ignition point is nearly independent of the mass and is therefore a material property for a given radiation intensity. The results of the fire technical mass comparison are shown in Table 3. A larger mass causes the HRR Peak, the MAHRE, the EHC and the fire duration to be increased. This effect was expected, as more material is available for the fire process and thus more heat can be released. As assumed, the time to ignition does not change with an increase in sample mass, since the same material is irradiated.

#### 3.1.3. Effects of Different Radiation Intensities on the Heat Release Rate of Polyamide

In this experiment, the effects of different radiation intensities on polyamide (PA 6) with a mass of 20 g were analyzed. The radiation intensities used, and the resulting fire parameters are shown in Table 4.

Figure 8 shows how the heat release of the material PA 6 changes as a function of the radiation intensity. Small irradiation intensity means late ignition time and low heat release, while with high irradiation intensity the material ignites quickly and also has a high heat release rate (HRR).

The key parameter for the ignition of solids is the surface temperature. If the irradiation intensity is low, it takes longer for the test sample to ignite. In the case of a radiation intensity of 25 kW/m^2^, it takes in the range of 9 min to ignite the material, in the case of 35 kW/m^2^, it takes only 3 min to ignite the material. With a higher intensity, the time to ignition is shorter (see Table 4).

The following equation is taken from the book of Quintiere [33] and describes the relationship between ignition time (t_ig_), irradiation intensity, ignition temperature (T_ig_), ambient temperature (T_ambient_) and various material constants:(1)$$t_{\mathrm{ig}}=C\left(k\rho c\right){\left[\frac{T_{\mathrm{ig}}-{\;T}_{\mathrm{ambient}}}{\mathrm{Radiation}\;\mathrm{intensity}\;}\right]}^{2}$$
t_ig_—time to ignition [s]T_ig_—ignition temperature [K]T_ambient_—ambient temperature [K]Radiation intensity [kW/m^2^]C(kρc)—Constant dependent on the material propertiesk—thermal conductivity [W/mK]ρ—density [kg/m^3^]c—heat capacity [J/K]

Figure 9 shows the ignition time as a function of the radiation intensity, whereby an exponential relationship can be recognized.

Polyamide was irradiated with irradiation intensities between 25 and 95 kW/m^2^ and Figure 10 shows that there is a linear relationship between irradiation intensity and the parameter $1/\sqrt{\mathrm{time}\;\mathrm{to}\;\mathrm{ignition}}$.

#### 3.1.4. Heat Release Rate from Different Low-Density Polyethylene (PE-LD) from Different Manufacturers (Borealis, Sabic, DOW)

The diagrams in Figure 11 shows the heat release rate of low-density PE (PE-LD) from different manufacturers. For the production of PE-LD, different catalysts and stabilizers are used by the individual companies, which may cause the different curve progressions of the HRR.

### 3.2. Flaming and Irradiation of Polypropylene (PP) with Various Flame Retardants

#### 3.2.1. Heat Release from PP-Copo with Different Filling Contents of Expandable Graphite

The improvement in the fire properties of PP homo through expandable graphite can be observed in Figure 12. An expandable graphite content of 5 and 10 weight-percent are most effective, considering the regarded range of expandable graphite. The material PP-Copo with 5% expandable graphite was also crosslinked beforehand during sample preparation with silane and peroxide.

#### 3.2.2. Effect of Expandable Graphite as a Flame Retardant

Expandable graphite is a very effective and versatile flame retardant. If a material with expandable graphite is heated, the graphite flakes expand to a multiple of the original volume, depending on the quality of the graphite. The expansion starts at temperatures above 180 °C. The expanded flakes have a “worm-like” appearance in the beginning and are usually several millimeters long (see Figure 13, Figure 14, Figure 15 and Figure 16).

Due to the layered structure of graphite, atoms or small molecules can be intercalated with acids between the carbon layers. During this process, a so-called expandable graphite salt or GIC (Graphite Intercalation Compound) is produced. Outstanding expandable graphite grades have a high proportion of intercalated layers. Usually, sulfur or nitrogen compounds are used as intercalation agents (see Figure 14) [34]. 

According to the information given by additive manufactures, expansion can commence at temperatures as low as 180 °C, depending on the material grade. The expansion and can occur suddenly and rapidly. In the case of free expansion, the final volume can be several hundred times greater than the initial volume [34].

The properties of expandable graphite, i.e., initial expansion temperature and degree of expansion, are primarily defined by the quality of intercalation (proportion of intercalated layers) and by the type of intercalation agent [34].

Next, the second fire retarding mechanism for expandable graphite is a carbon donation in combination with polyols. This results in a protective layer due to charring, if an acid donor (e.g., ammonium phosphate) and a blowing agent (e.g., melamine) are added [33,34,35].

In the following pictures (in Figure 15), the growth of expandable graphite is visible. The expansion of a PP filled with 5 wt % expandable graphite can be observed. The images show the development of the expandable graphite layer over time during irradiation with the cone calorimeter.

The pictures of Figure 16 show electron microscope images with different magnifications of the expanded graphite [36].

#### 3.2.3. Heat Release Rate and Burning Behavior (Schlyter Test) of PP with Various Flame Retardants

As part of Hannelore Mattausch’s research work about the development of halogen-free flame retardant polypropylene compounds, a large number of PP test samples with various flame retardants were analyzed for fire protection purposes [37,38,39,40]. PP with various flame retardants was produced and the test sample version was also exposed to a flame for a complete fire-related assessment. The results of the flame tests are presented first because they require more extensive explanations than the cone calorimeter tests.

The following samples have been tested for low flammability according to the Schlyter test. The tests at hand could not be carried out in accordance with the Austrian standard, because only plates of smaller size were available. The standardized sample size would be 160 mm × 160 mm with a thickness of 4 mm. However, since all tests were carried out in the same way, the results are comparable with one another. The burner of the Schlyter test shown in Figure 17 is 17 cm wide and has a total burner output of 1700 Watt, i.e., 100 Watt/cm. A flame was applied to approximately 8 cm test specimens, and a burner output of approximately 800 watts was thus supplied. In order to simulate a strong flame load, the Schlyter test with the very small test sample sizes was used to rank the various flame retardants.

Figure 17 shows the flame exposure of sample PP+5%Gra + 5%AlPh + 5%MMT. The designation Gra means expandable graphite. Figure 18 shows the comparison of the burned lengths from samples TGM 1 to TGM 6.

Figure 18 depicts the first six samples after performing the Schlyter test. Specimens TGM 4 and TGM 5 are burned over their entire length (80 cm), sample TGM 6 over 76 cm, while samples TGM 1 to TGM 3 are burned up to a maximum length of 25 cm.

Figure 19 shows the MARHE values and by colouring the bars the length burned in the Schlyter test. The burned length is classified, and the respective bar color symbolizes the burned length. In the tests shown, it is shown how some flame retardants have a favourable effect on irradiation and others on flame.

Figure 20 represents those materials that show favorable values in terms of flame resistance and heat release when irradiated. Essentially, it can be said that expandable graphite (abbreviation Gra in Figure 19 and Figure 20) is an effective flame retardant for PP; additionally, some additives can further reduce heat release. In general, it cannot be said which additives reduce the heat release and also have a beneficial effect in terms of fire behavior. It was also found that multiple flame retardants such as montmorillonite (MMT), aluminum polyphosphate (AlPh) and zeolite (Zeolith) can significantly worsen the fire behavior. Detailed investigations must therefore always be carried out in this regard.

#### 3.2.4. Change in the Heat Release rate of PP Sheets Due to Various Additivation of the Test Samples and Results of the UL 94-HB Test

Fire properties of polypropylene (PP) with various additives determined with two radiation intensities (35 kW/m^2^ and 50 kW/m^2^) and with flaming according UL 94 horizontal burning (UL 94 HB) are analyzed. The maximum average rate of heat emission (MARHE) was used to describe the fire behavior regarding the radiation intensity.

With irradiation by the cone calorimeter and flame treatment according to the UL 94 HB test, virgin PP and PP treated with crosslinking agents, flame retardants and synergists were tested. Table 5 contains all measurement results in detail and in Figure 21 the MARHE for the two irradiation intensities and the UL 94 HB result are visualized. Table 5 also includes HRRpeak (written as HRRp), but these values are only included for comparison with the MARHE value. PP1 and PP2 only indicate different PP types.

Furthermore, samples AM 11 (PP1 + FR4 12.5%) and AM 12 (PP1 + FR5 6%) do not have a fire classification at UL 94 HB. This means that AM 11 and AM 12 delivered the best results in the UL 94 HB test, which is described in Table 5 by the burning rate of 0. Sample AM 06 failed this test because the burning rate is greater than 75 mm/min. These would have to be tested with UL 94 V in order to achieve a result.

The results of Figure 21 and Table 5 indicate that, with PP, the heat release rate HRR and also the fire behavior is influenced very positively by a crosslinking agent and a flame retardant. The synergist did not have any positive effect in the investigations presented.

The synergist NOR is a monomeric N-alkoxy hindered amine (NOR HAS) which acts as a flame retardant in polyolefin applications. Very small amounts of this additives are beneficial for activating the flame-retardant effect and for reducing the overall amount of flame retardants in the formulation. The flame retardants, mentioned in Table 5, are different phosphorous based flame retardants.

Figure 22 shows the comparison of pure PP (AM00) and the effect of crosslinker (AM01) and crosslinker and flame retardant FR1 (AM02). The increasing number of additives lower the heat release. It should be noted, however, that the crosslinker increases the burn rate from 36 mm/min to 50 mm/min and only the flame retardant FR1 can decisively reduce the burn rate to 20 mm/min (see Table 5). This may be caused due to the stable chemical bondings caused by the crosslinking.

Figure 23 compares the time dependent HRR for the materials with the most effective flame retardant additivation. Figure 23 shows that the maximum heat release rate of PP (AM00), at radiation intensity of 35 kW/m^2^ and 50 kW/m^2^, is significantly reduced by the crosslinker and 10% FR3 and 0.8% synergist (material AM13). The material (AM13) has the lowest heat release rate, but the burning rate in the UL94 test is higher than in other cases (see Figure 21).

In contrast, it can be clearly seen that the addition of the synergist has a negative effect on the fire properties. Therefore, it should not be used unless it is necessary for other reasons. With a crosslinker content of 1.3%, Flame Retardant FR3 shows the best effect both at 35 kW/m^2^ and at 50 kW/m^2^ irradiation, as this reduces the MARHE most effectively related to the virgin material.

## 4. Conclusions

Cone experiments were mainly performed to determine the heat release rate of different polymers when irradiating the test samples. The heat release of polymeric materials without flame retardants was investigated. Pressed PP granules were tested from day 1 to day 99 for heat release. PP produced by injection molding was also tested on day 1 and day 33. The mass in both cases was slightly different and therefore the results cannot be directly compared, but in all cases changes in heat release were observed, especially in the initial phase (see Figure 3 and Figure 5). It was determined with DSC examination that the differences in the burning behavior of PP will not be caused by post-crystallization.

Polyamide was irradiated with irradiation intensities between 25 kW/m^2^ and 95 kW/m^2^, and it was shown that there is a linear relationship between irradiation intensity and the parameter 1/$\sqrt{\mathrm{time}\;\mathrm{to}\;\mathrm{ignition}}$ (Figure 10). The heat release rates of PE-LD from different manufacturers showed a slightly different behavior (Figure 11).

For PP-Copo, expandable graphite is an effective flame retardant for irradiation, whereby with a proportion of 5% the heat release can be strongly reduced and a proportion of 10% only brings small further improvements (Figure 12). The effect of expandable graphite is clearly illustrated in Figure 13 and Figure 15. Expandable graphite is a very good flame retardant. It is only with materials that must also meet optical requirements that the visible black small expandable graphite inclusions could be disturbing.

PP was tested with different flame retardants not only with regard to heat release, but also with flames of varying intensity. The flame treatment according to the UL 94 test is carried out with 50 W flame power. In order to also test PP with different flame retardants with a very strong flame, the Schlyter test was chosen. The entire burner has an output of 1700 W, but since the plates were only 16 cm wide instead of 30 cm (see Figure 17), it can be assumed that a burner output of approximately 800 watts was fed into the test samples.

In the case of PP, it is proven that 5% expandable graphite is also very effective as a flame retardant with regard to flaming, as well as to irradiation. When 5% expandable graphite and 5% AlPh are added, the burning behavior remains approximately the same and is not further improved. Figure 20 and Figure 22 show the best materials in terms of low heat release and when flaming with the Schlyter test with high flame output.

The UL 94 V test allows only a passed or failed classification and therefore the UL 94 HB test was performed. In the UL 94 HB test, the burning rate is indicated and is a finer criterion for material evaluation. The results of Figure 21 and Table 5 indicate that, with PP, the heat release rate (HRR) and also the fire behavior are influenced very positively by a crosslinking agent and a flame retardant. The synergist did not have any positive effect in the investigations presented.

Table 5 and Figure 22 show that a 1.3% crosslinker (AM01) reduces the heat release somewhat, but increases the rate of combustion, i.e., deteriorates the properties during flame exposure. The additional addition of 10% of the flame retardant FR1 (material AM02) slightly reduces the heat release and significantly lowers the burning rate. Figure 23 shows the favorable properties of material (AM13) with 1.3% crosslinker, 0.8% synergist and 10% FR3.

## Figures and Tables

**Figure 1 materials-13-05756-f001:**
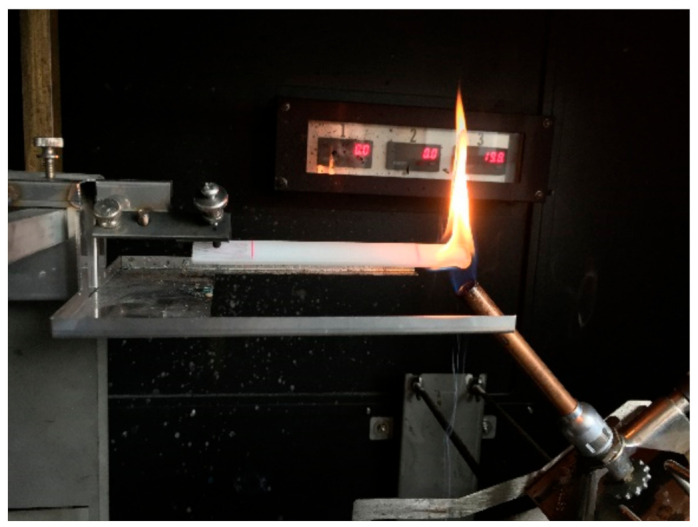
Flame treatment in accordance with UL 94 HB.

**Figure 2 materials-13-05756-f002:**
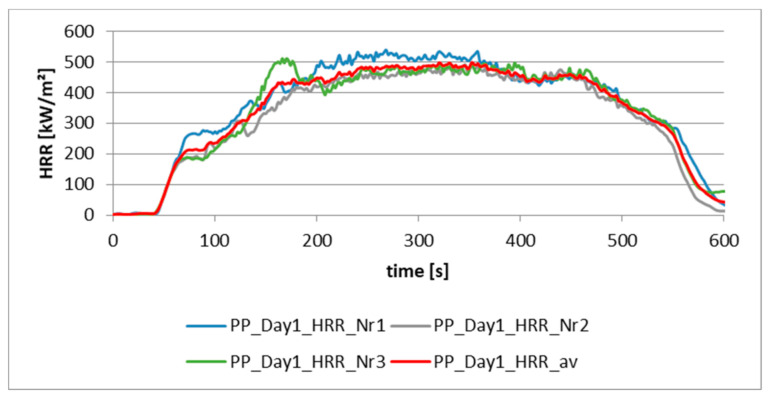
Heat release rate (HRR) of three samples of PP (granules were pressed, m = 50 g) on the first day after production.

**Figure 3 materials-13-05756-f003:**
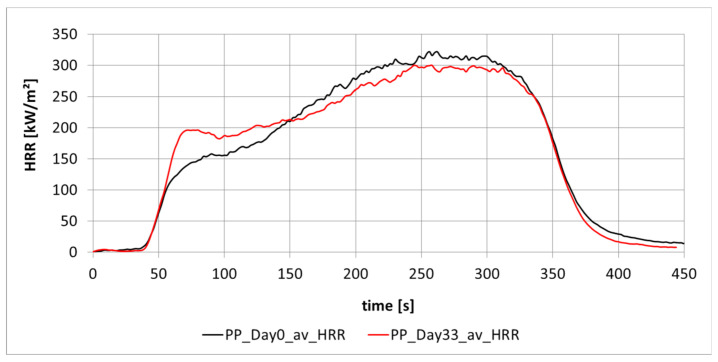
Heat release of PP (m = 20 g) produced by injection molding measured on the day of production and 33 days later.

**Figure 4 materials-13-05756-f004:**
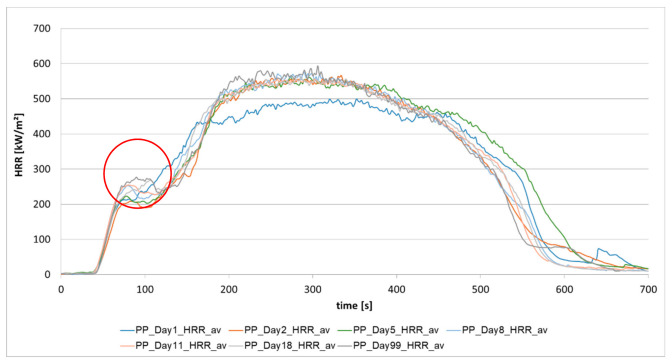
Heat Release rate of pressed granules of PP (m = 50 g) from the first day to day 99.

**Figure 5 materials-13-05756-f005:**
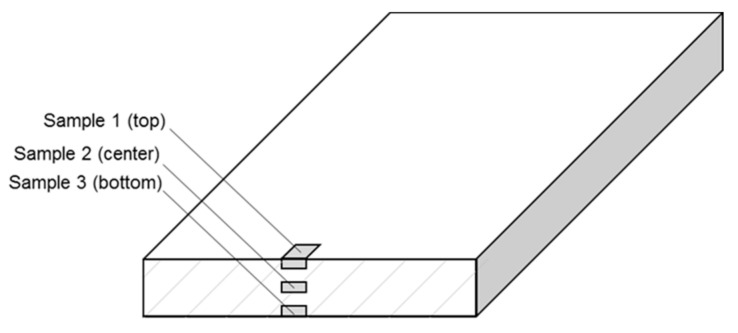
Positions of DSC-samples over cross section.

**Figure 6 materials-13-05756-f006:**
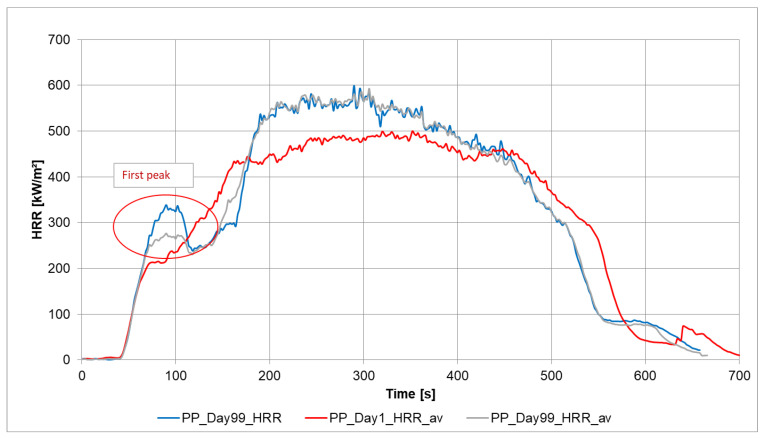
Heat Release rate of pressed granules of PP from the first day (average) and day 99 (one measurement (PP_Day99_HRR) and average of three measurements PP_Day99_HRR_av).

**Figure 7 materials-13-05756-f007:**
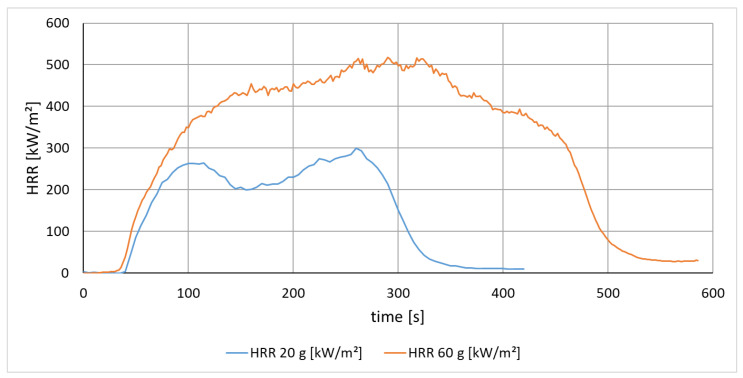
Comparison of the HRR of PA 6 with 20 g and 60 g (radiation intensity 50 kW/m^2^).

**Figure 8 materials-13-05756-f008:**
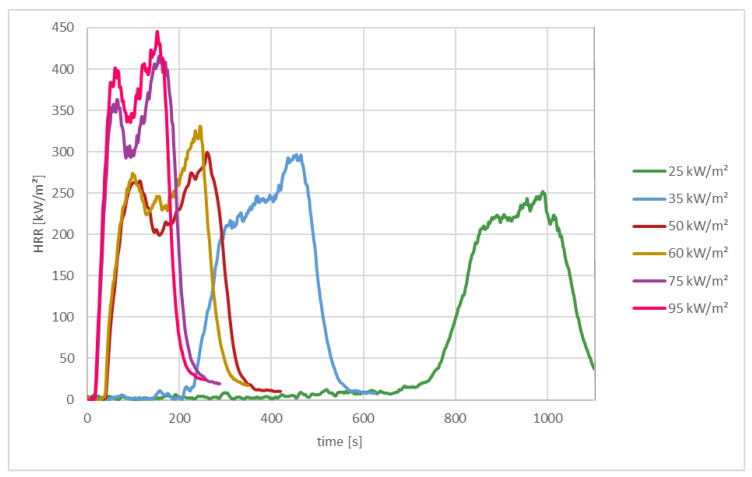
Heat release rate of PA 6 at different radiation intensities.

**Figure 9 materials-13-05756-f009:**
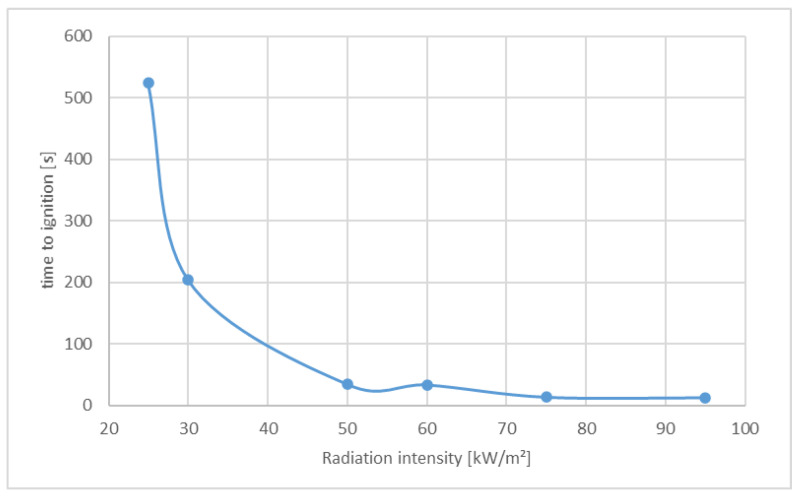
Time to ignition dependent on radiation intensity.

**Figure 10 materials-13-05756-f010:**
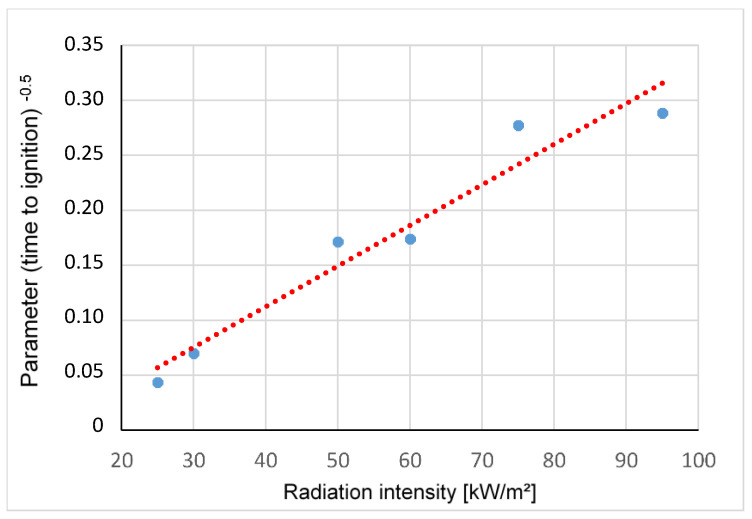
Linear relationship between the different radiation intensities.

**Figure 11 materials-13-05756-f011:**
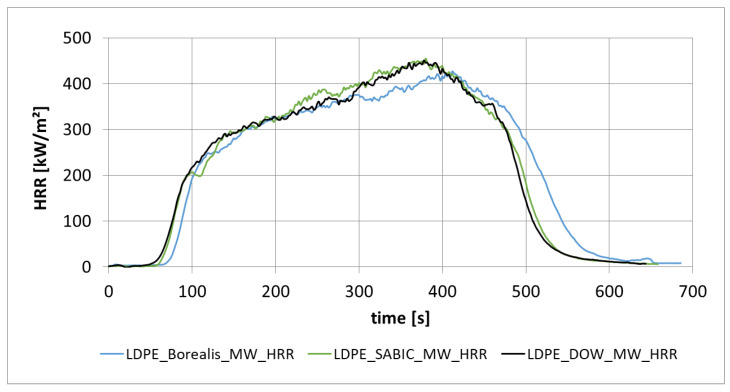
Different heat release rate from different PE-LD manufacturers (Borealis, Sabic, Dow).

**Figure 12 materials-13-05756-f012:**
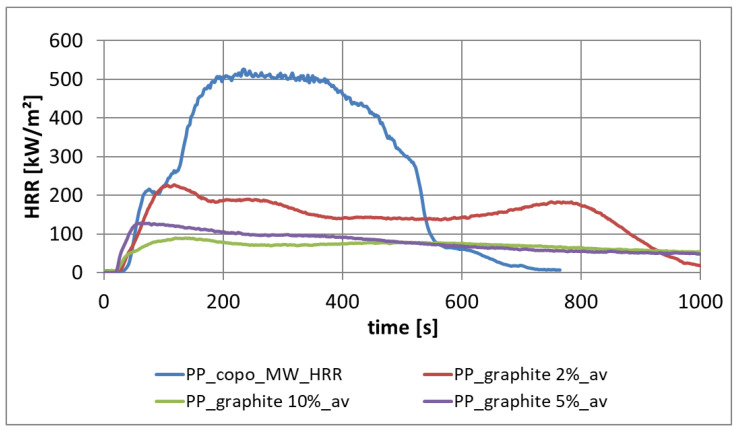
Heat release rate of PP-Copo with different portions of expandable graphite.

**Figure 13 materials-13-05756-f013:**
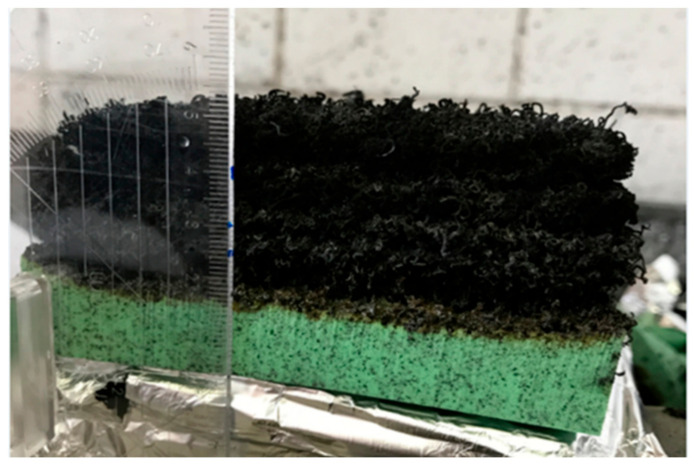
About 3 cm high expanded graphite layer after irradiation of a foam with a thickness of 1.5 cm.

**Figure 14 materials-13-05756-f014:**
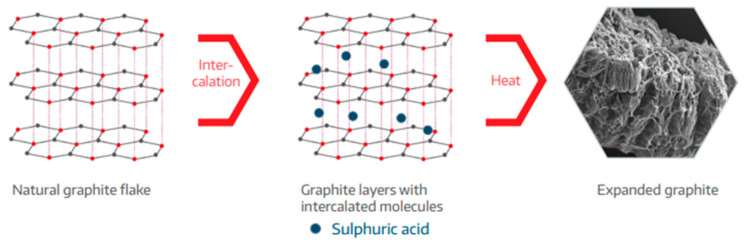
Intercalation of expandable graphite [34], reproduced with permission from Graphit Kropfmuehl GmbH, Hauzenberg.

**Figure 15 materials-13-05756-f015:**
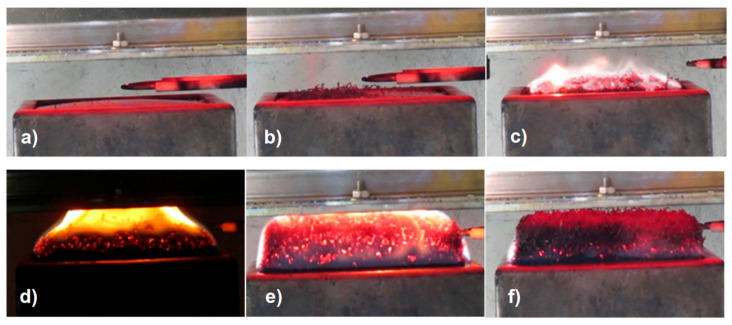
Imagines showing the growth of expandable graphite: (**a**) before ignition, (**b**) expandable graphite begins to expand ’worm like’, (**c**) strong foaming and start of burning, (**d**) expanding more and more, (**e**) maximum foam height, (**f**) glowing of expandable graphite has ended.

**Figure 16 materials-13-05756-f016:**
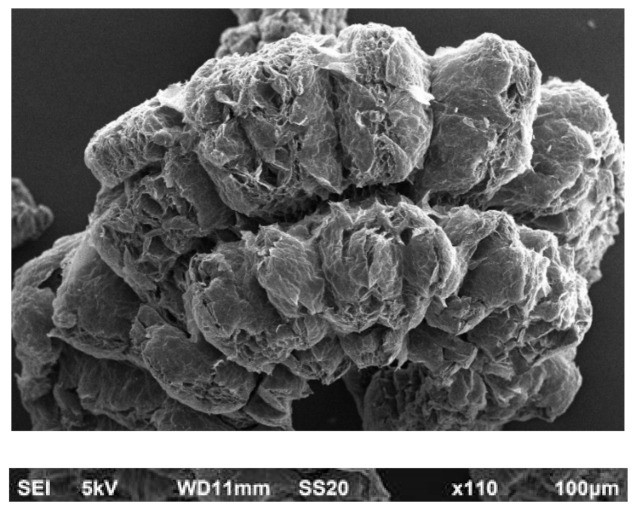
Scanning electron microscope images of expanded graphite, reproduced with permission from Graphit Kropfmuehl GmbH, Hauzenberg.

**Figure 17 materials-13-05756-f017:**
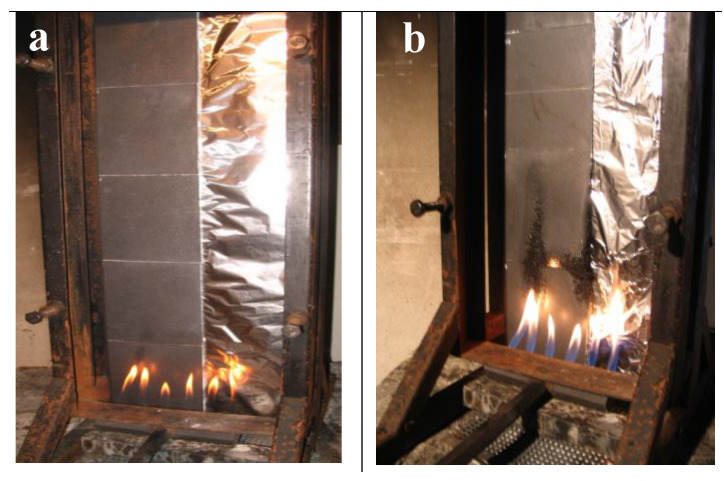
Example (sample PP + 5%Gra + 5%AlPh + 5%MMT) of a flame treatment based on the Schlyter test at different times after the start (**a**) and after few minutes (**b**) of the flame treatment.

**Figure 18 materials-13-05756-f018:**
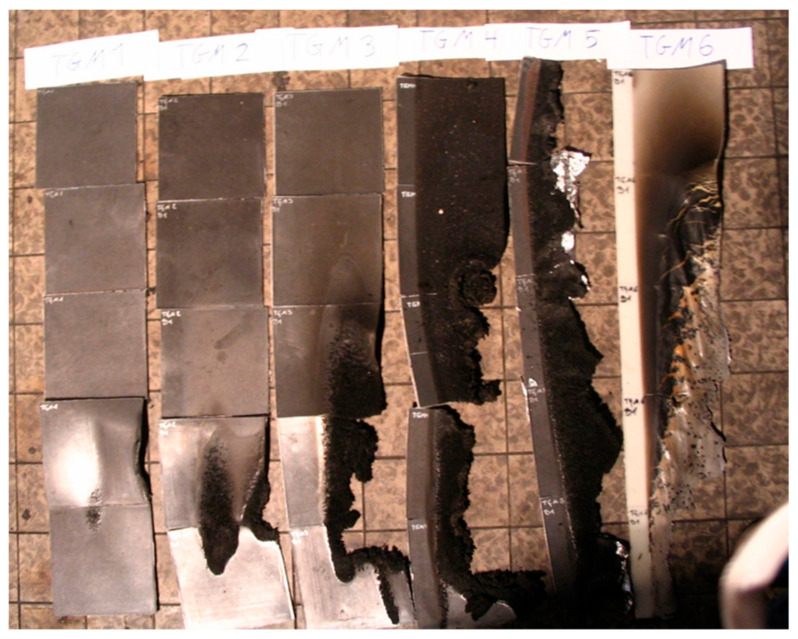
Test samples TGM 1 to TGM 6 after carrying out the Schlyter test. (TGM1: PP + 5%Gra, TGM2: PP + 5%Gra + 5%AlPh, TGM3: PP + 5%Gra + 5%MMT, TGM4: PP + 5%Gra + 5%AlPh + 5%MMT, TGM5: PP + 5%Gra + 5%MgOH, TGM6: PP + 5%MgOH).

**Figure 19 materials-13-05756-f019:**
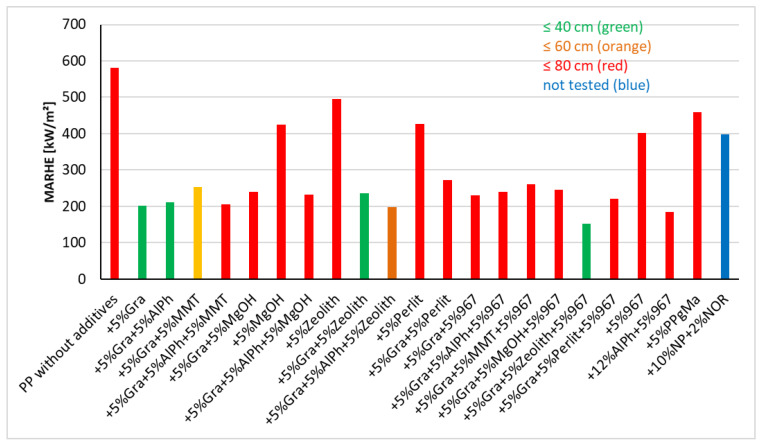
Comparison of the cone results (MARHE) with the burned lengths according to Schlyter test (shown in colour) for additivated PP.

**Figure 20 materials-13-05756-f020:**
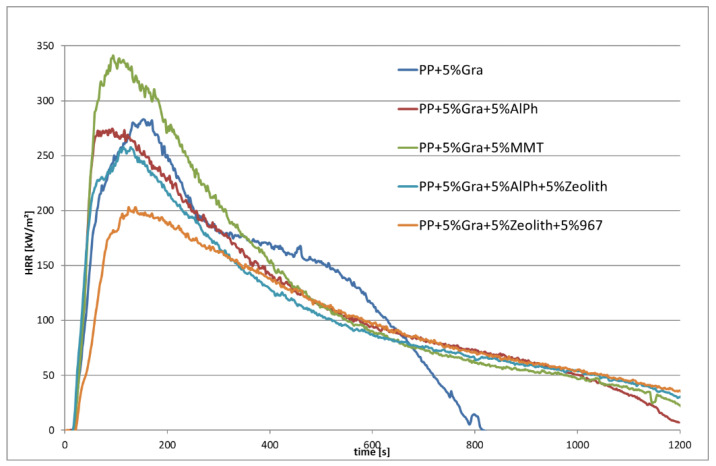
Best materials concerning Heat Release Rate (HRR) and burned length of Schlyter test.

**Figure 21 materials-13-05756-f021:**
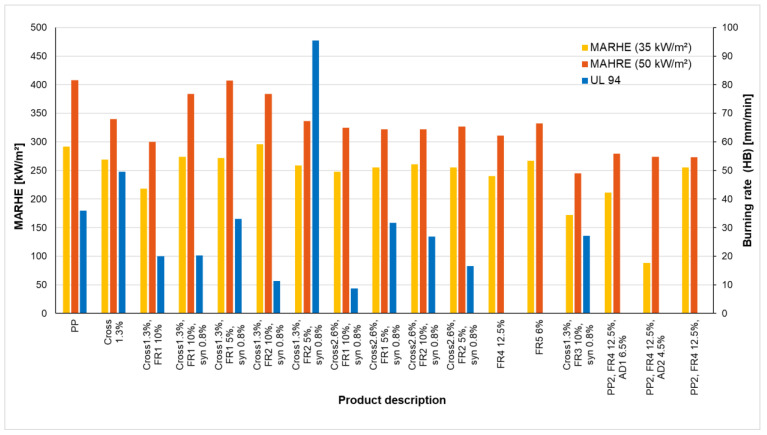
Fire properties of PP with various additives determined with irradiation (result MARHE) and with (horizontal) burning rate (UL94 HB).

**Figure 22 materials-13-05756-f022:**
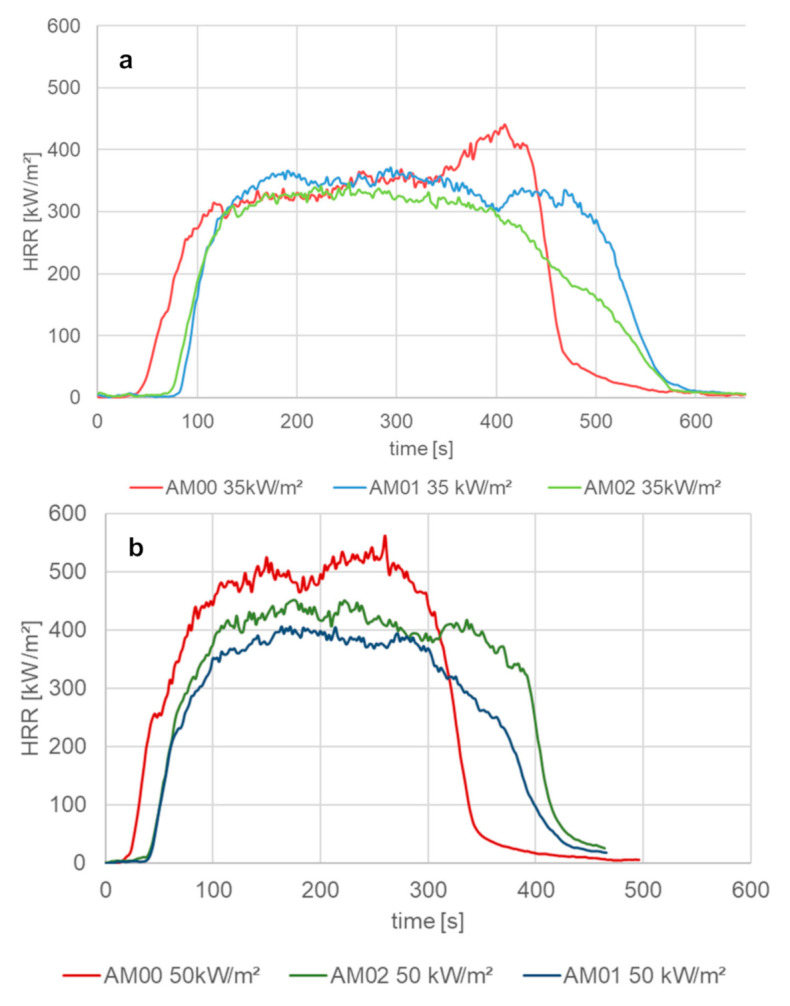
The heat release rate of PP (AM00) and PP with a crosslinker (AM01 with PP + cross 1.3%) and a flame retardant (AM02) with PP + cross 1.3% + FR1 10%) at radiation intensity of 35 kW/m^2^ (**a**) and 50 kW/m^2^ (**b**).

**Figure 23 materials-13-05756-f023:**
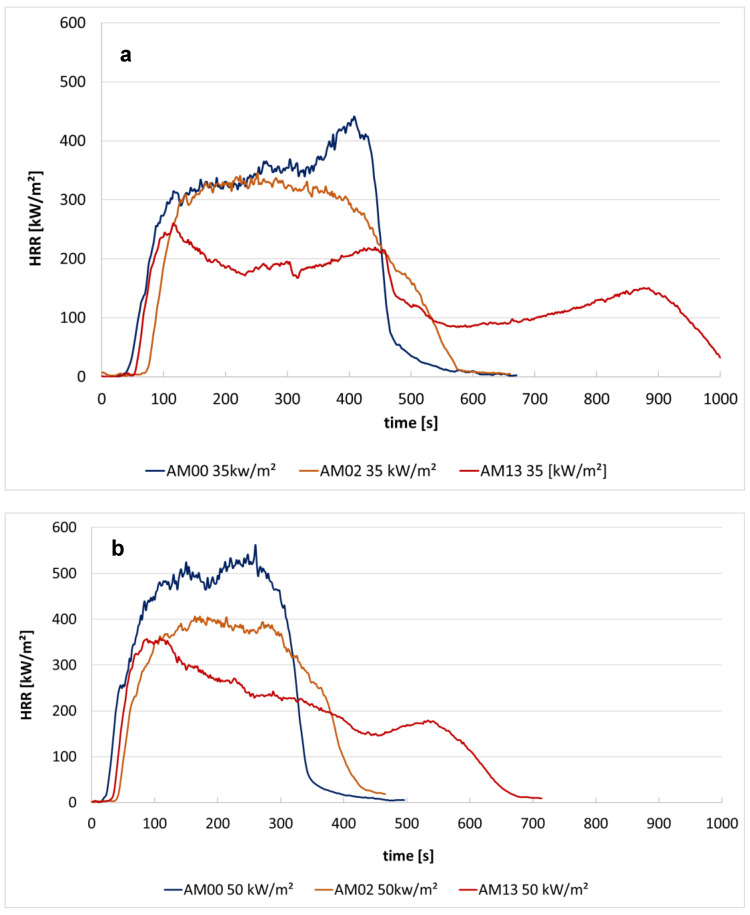
The heat release rate of PP (AM00) and PP with a crosslinker and a flame retardant (AM02 with PP+cross 1.3% + FR1 10% and AM13 with PP + cross 1.3% + FR3 10% + syn 0.8%) at radiation intensity of 35 kW/m^2^ (**a**) and 50 kW/m^2^ (**b**).

**Table 1 materials-13-05756-t001:** Specification of materials.

Material	Producer	Polymer Type	Grade	Additional Information
PP	Borealis	PP Copolymer	BB412E	medium molecular weight block copolymer
PA	BASF	PA 6	Ultramid B27E	low viscosity general-purpose, extrusion grade
PE-LD	Borealis	PE low density	FT5230	unmodified for film extrusion
PE-LD	Dow	PE low density	300E/302E	unmodified for blow film extrusion
PE-LD	Sabic	PE low density	2601 × 1	ultra melt strength grade with slip and antiblocking agents for foam applications
Gra	AMG Mining	-	ES350F5	Expandable graphite, D_80_ 300 µm
Zeolite	Paltentaler Minerals	-	100_15	D_50_ 15 μm
MMT	Rockwood Clay Additives	-	Nanofil 5	Montmorillonite, D_50_ 10 μm
MgOH	Ankerpoort N.V.	-	Securoc B9	Magnesium hydroxide, D_50_ 2.6 µm
Silane	Sigma Aldrich	-	3-(trimethoxysilyl) propyl methacrylate	-
Peroxide	Sigma Aldrich	-	Dicumyl peroxide	-
AlPh	Italmatch Chemicals SpA	-	Phoslite B85AX	Aluminium phosphinate
Perlit	Montanuniversitaet Leoben, Chair of Mineral Processing	-	-	Expanded perlite, D_50_ 0.525 mm, bublon process
967	Thor	-	AFLAMMIT PPN 967	Multi-component blend based on ammonium polyphosphate
NOR	BASF	-	NOR116	monomeric N-alkoxy hindered amine (triazine derivative)
NP	Adeka Palmarole	-	ADK STAB FP-2100JC	nitrogen/phosphorus-based, halogen-free flame retardant

**Table 2 materials-13-05756-t002:** Degree of Crystallinity of PP after conditioning at various parameters.

Storing Time [Days]	Storing Temperature [°C]	Degree of Crystallinity [%]
1	23	34
5	8	35
7	23	34
5	80	35

**Table 3 materials-13-05756-t003:** Comparison of combustion parameters of PA 6 with different mass.

Parameter	Sample Mass [g]	
	20	60
HRR Peak [kW/m^2^]	299	530
MARHE [kW/m^2^]	197	375
EHC [MJ/kg]	29	77
Time of Ignition [s]	34	32
Burning Time [s]	266	552

**Table 4 materials-13-05756-t004:** Combustion parameter of PA 6 yield with different radiation intensity.

Radiation Intensity [kW/m^2^]	25	35	50	60	75	95
HRR Peak [kW/m^2^]	252	297	299	330	415	445
MAHRE [kW/m^2^]	57	116	197	205	291	313
EHC [MJ/kg]	32	31	29	30	29	30
Time to Ignition t*_ig_* [s]	524	204	34	33	13	12
Burning Time [s]	626	326	266	348	256	218

**Table 5 materials-13-05756-t005:** Fire properties (HRRpeak and MARHE) of PP with various additives for radiation intensity 35 and 50 kW/m^2^ and horizontal burning (UL 94 HB) with result burning length.

		Radiation Int. 35 kW/m^2^		Radiation Int. 50 kW/m^2^		UL94 HB
Sample	Weight Percentages of Additives [%]	HRRp	MARHE	HRRp	MARHE	mm/min
AM 00	PP1 100% (native)	441	292	562	408	35.9
AM 01	PP1 + Cross 1.3%	371	269	451	340	49.6
AM 02	PP1 + Cross 1.3% + FR1 10%	346	218	406	300	20
AM 03	PP1 + Cross 1.3% + FR1 10% + Syn 0.8%	398	274	593	384	20.3
AM 04	PP1 + Cross 1.3% + FR1 5% + Syn 0.8%	377	272	605	407	33
AM 05	PP1 +Cross 1.3% + FR2 10% + Syn 0.8%	435	296	589	384	11.4
AM 06	PP1 + Cross 1.3% + FR2 5% + Syn 0.8%	399	259	534	336	95.5
AM 07	PP1 + Cross 2.6% + FR1 10% + Syn 0.8%	375	248	478	325	8.7
AM 08	PP1 + Cross 2.6% + FR1 5% + Syn 0.8%	379	255	443	322	31.7
AM 09	PP1 + Cross 2.6% + FR2 10% + Syn 0.8%	374	261	443	322	26.8
AM 10	PP1 + Cross 2.6% + FR2 5% + Syn 0.8%	346	255	415	327	16.5
AM 11	PP1 + FR4 12.5%	393	240	483	311	0
AM 12	PP1 + FR5 6%	536	267	584	332	0
AM 13	PP1 + Cross 1.3% + FR3 10% + Syn 0.8%	261	172	358	245	27.2
RO 01	PP2 + FR4 12.5% + Ad1 6.5%	334	211	387	279	–
RO 02	PP2 + FR4 12.5% + Ad2 4.5%	136	88	414	274	–
RO 03	PP2 + FR4 12.5%	381	255	435	273	–

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
