# Peer review of "Analysis of the Fire Behavior of Polymers (PP, PA 6 and PE-LD) and Their Improvement Using Various Flame Retardants"

_materials, 2020, doi:10.3390/ma13245756_

Round 1

Reviewer 1 Report

The paper is very practical in the content and suitable for publication after the authors revised the manuscript.

  1. On line 65, the authors should provide the basic information of the experimental materials, PP, PE, EG, flame retardants and polyamide, for example, producer, molecular weight, chemical structure....
  2. On line 165, the difference is very small from Table 1. The authors should explain it in detail.
  3. In the caption of figure 2, the expandable graphite and expanded graphite is different. The authors should use the correct words.
  4. In figure 20, what are BG, AlPh, MMT and Zeolith? The authors should introduce them in the text.
  5. On line 356, what is the synergist? The authors should provide the discussion about the effect.
  6. On line 357, what are the cross linkers? Please provide some discussions about the effect.
  7. The section of conclusions is too long. The authors should rewrite it. 

Author Response

On line 65, the authors should provide the basic information of the experimental materials, ....

Information, as far as available, about the materials and additives will be added in an new table.

On line 165, the difference is very small from Table 1. The authors should explain it in detail.

We are not sure about your comment, but we have referenced our results of the burning behaviour to the DSC-results. Further, we added assumptions what may cause the change in the burning characteristics.

In the caption of figure 2, the expandable graphite and expanded graphite is different. The authors should use the correct words.

We guess you meant Fig. 14? We have unified the terms to expanded graphite.

In figure 20, what are BG, AlPh, MMT and Zeolith? The authors should introduce them in the text.

We added explanations to the material description and added the abbreviations in the text too.

On line 356, what is the synergist? The authors should provide the discussion about the effect.

We added some details.

On line 357, what are the cross linkers? Please provide some discussions about the effect.

We added some details and an explanation, why the burning behaviour was effected.

The section of conclusions is too long. The authors should rewrite it

We tried to shorten it.

We would like to thank you for your helpful comments.

Reviewer 2 Report

The particular type of polypropylene as well as polyamides have to be indicated in the title as well as in the Abstract. Specific flame retardant additives also need to be indicated in the Abstract

The keywords include: “polyethylene” and “Cross-linking „ which are not mentioned in the Abstract

The introduction is rather simplistic. There is neither information if any other researchers have analysed the studied properties, nor which polymers were analysed by applying  the mentioned technique.

Information concerning materials is non-existent. No information whatsoever on the  injection moulding machine as well as temperature of injection has been provided. Moreover, the type of cone calorimeter has to be specified.

Polyamide is a term describing a wide spectrum of polymers. The Authors have to clearly indicate the type of polyamide as well as the source from which these the polymers in question have been obtained. 

There is no explanation why the changes in burning behaviour of polypropylene  were analysed starting from the day after production and ending with the  99th day of storage at room temperature.

Section 3.1.1, referring to aging of polypropylene (PP) - Lines 124-127 - is a description of methodology, not results.

Authors claim that “Above all, the different heat release rates around the second 70 indicate a slight material change in the test sample, which is probably caused by aging.” The reason for these  differences to occur has to be established. I suggest applying the FTIR and DSC techniques in an aim to analyse if aging of PP has actually taken place.

While discussing results the Authors have omitted Figure 4, proceeding from Figure 3 directly to Figure 5.

Comparison of samples stored at room temperature with the sample stored at 80 oC does not seem reasonable, and needs to be justified in some way. Why have the samples stored at room temperature not been analysed?

In the methodology section of the paper there is no information about tempered samples. The work is overwhelmingly incoherent In the section relating to materials both methods as well as the description of results have been included.

It is impossible to compare the results obtained in relation to the PP samples when:

  • FIGURE 2 depicts results recorded in the case of PP samples  in the shape of pressed granules on the first day after production; no information about the mass (20 or 50g?) has been provided
  • FIGURE 3 indicates the heat release of PP samples obtained in the injection moulding process (mass equals 20 g) with results recorded on the day of production and 33 days later
  • Figure 5. depicts the heat release rate of pressed granules of PP (mass equals 50 g) after 1; 2; 5; 8; 11; 18; 99 days!

In the 3.1.2 section –(page 6) the Authors proceed directly to discussing polyamide with no mention of a particular PA type.

In the 3.1.3 section (page 8) Effects of different radiation intensities on the heat release rate of polyamide were analysed. The same effects should be discussed in relation to PP samples.

In the 3.1.4 section (page 10) Authors describe the heat release rate of different, low-density polyethylene (PE-LD) from different manufacturers. Neither in the Abstract part of the work nor in the Introduction, nor the Materials sections is there any information on the analysis of LDPE.

In the 3.2 section, the Authors present the flaming and irradiation procedures applied to polypropylene (PP) containing various flame retardants. There is no indication as to the type of flame retardants and no description of the composition and methodology of obtaining the PP-based materials.

The Authors claim that “An expandable graphite content of 5 and 10 wt% are most effective.” (line 245). There is no evidence that the effective content of expandable graphite has actually been determined.

There is no explanation of what “PP-Homo” means.

Figure 16 presents SEM images which are taken from another publication. It has to be clarified if the Authors have permission to copy the SEM images.

 Figure 19 presents results relating to samples whose composition, materials, methodology and the abbreviated designations make it impossible to analyze and validate these results. It is impossible to discuss the results obtained in the case of materials we know nothing about.

I must regretfully admit that rarely do I see such inadequate and deficient publications. The abstract and introduction sections provide no viable information on the materials used in the study. The methodology section is incoherent and inconsistent, lacking most basic information that should normally be provided in relation to materials and fire retardants that were used in the study. It is impossible to decipher the abbreviated designations of the samples. Moreover, the results and their discussion are inconsistent to an extent that disqualifies the submitted work for publication. I am forced to insist on an immediate rejection of this paper.

Author Response

ad particular type of PP, PA
We added additional information about the used materials.
Adding details in the abstract would exceed the recommended length of the text.

ad keywords
The keywords were added to the abstract.

ad Simplistic introduction
Further information were provided.

ad Material information / PA-type
Further information were provided and descriptions added.

ad information about injection moulding
Data were added and the machine type is now mentioned.

ad information about cone calorimeter
The producer and production year are given in the text. It is mentioned, that the required standard is fulfilled with this machine. There are no further machine types commonly used in Central Europe. Next, there are no further information given by the producer.

ad explanation for storing PP
It is commonly known that PP is sensitive for aging at room temperature. Therefore, we thought it would be a good idea to test the fire behaviour depending on the storage time.

ad section 3.1.1
The parameters were stated here since it is useful in this content and helpful to understand the results. It seemed useful to give some details of the measurement results at this point to make the text easier to read.

ad change by aging
It was not the focus of this work to analyse aging of PP in detail. It was only the intention to verify if there is a change in fire behaviour detectable by using cone calorimeter. Since the influence was only small, further tests were not carried out.

ad omitting Fig. 4 in discussion
Since Fig. 4 shows only how DSC-samples were prepared it seems not to be important to refer on this figure in the discussion.

ad reason for analysing PP at 80 °C
As it was mentioned "It was supposed that the crystallinity would rise due to the tempering." Influence of the crystallinity on the burning behaviour was part of our research focus and therefore the stated tests were carried out.

ad no analysing of PP stored at room temperature
All burning tests (e. g. Fig. 5 and 6) were done using PP that was stored at room temperature. We are not sure, what you mean.

Tempering at 80 °C was only done for investigation of the maximum achievable change in crystallinity, since we assumed a change in crystallinity after storing at room temperature for 99 days.

ad incoherency
We thought it would be best to describe the principles of our measurements in the section "methods" and detail the correlating works (e. g. tempering) in the results. This seems suitable to us as it enhances the reading and understanding.

ad impossible comparison (Fig. 2 and 3)
The mass, which was stated in the text, was added in the description of figure 2. 

ad missing PA-type in section 3.1.3
The PA-type (PA 6) was added.

ad different radiation intensity for PP
The influence of the intensity was only discussed for PA as an example. It was the intension to indicate differences of the burning behaviour and their influence factors in general. Hence, not all influences (pressing vs. injection moulding, different radiation intensities, additivation with flame retardants, etc.) was determined for each and every material.
This would exceed the useful content of this paper.

ad missing information for PE-LD
Material information were added and the analysis of PE-LD is now mentioned in the abstract.

ad section 3.2 (PP)
Information about the flame retardants and the polymer were added in chapter Materials and Methods.

ad effective graphite content
It is now made clear that most effective means in regard to the analysed range.

ad PP-Homo
A short explanation is now added in chapter materials and methods.

ad Fig. 16
We changed figure 16 and have now used a picture, we have the permission for.

ad Fig. 19
We added the meaning of the abbreviations of the flame retardants in the text.

Thanks for your comments.

Reviewer 3 Report

This paper reported analysis of the fire behavior of polymers and improvement using various flame retardants. The paper has been written well. However, certain concerns need to be noticed and clarified.

  1. Introduction is too short and must be improved highlighting the aim of the paper. Some research background should be explained and the research content of others should be introduced.
  2. In the materials and methods section, add some subheadings as appropriate to make the article appear more organized.
  3. Please use more words in Figure 3 to make them clear for readers.
  4. It is best not to use other people's references in the conclusion section. It should be a summary of what you have studied.

Author Response

ad 1) The following additions were made:

It was the aim to investigate the influence of different parameters, e. g. the aging of polypropylene, the influence of the radiation intensity and of various flame retardants. Further, different polymers were tested to give an overview about their different burning behaviour.
It has to be stated that not all variations were determined for each polymer as this would exceed the scope of this paper.

The given data were obtained during various academic projects at the Montanuniversitaet Leoben (Technical University of Leoben, Austria) and the Laboratory for Polymer Engineering (LKT, Vienna, Austria) in cooperation with the plastics technology department of the technical college TGM at Vienna. The focus of these projects was to determine the basic fire behaviour of polymers.

ad 2) The chapter was structured by subheadings (Sample preparation, Determination of the fire behaviour, Precision of fire behaviour tests, Differential Scanning Calorimetry).

ad 3) We tried to improve our description (see below).

The change of the characteristic of the curve progression is clearly visible due to a local maximum of the red curve (indicating the samples after 33 days) at about 70 seconds after start of measurement.

ad 4) We removed the references with the whole last paragraph and moved it to the chapter materials and measurements.

We hope we made our changes like you suggested it. We want to thank you for your comments and for your helpful feedback.

Round 2

Reviewer 2 Report

  1. The designations of the analyzed polymers have to be indicated in the title of the manuscript. Otherwise, the reader can be misled into thinking that all types of polymers have been studied.
  2. The introduction provides no viable information on the materials used in the study. A review of literature, devoted to the properties of flame retardants used in the research has to be included.
  3. More information related to the mineral additives applied in the research is required. For example, the Authors reported having used MMT – Nanofil5. It is well known that Nanofil 5 is a modified variant of MMT. Taking into account that a modifier significantly influences the structure and properties of materials, modified and unmodified samples have to be analyzed and compared. Moreover, the name and the structure of the modifying agent has to be described.
  4. In the 2.1. section (Sample preparation) there is no indication of conditions (temperature, pressure) in which the formation of all types of composites (PE, PP, and PA) had occurred. Moreover, the designations and the composition of all of the obtained materials have to be presented in a table.
  5. It is well known that the addition of a filler in the form of MMT or zeolite causes the formation of different types of composites: exfoliated, intercalated, or simply phase separated. The structure significantly influences the flammability of the obtained composites, and for this reason, the structure has to be established by means of the XRD or TEM techniques.
  6. Figure 3 – the same chart is presented twice. The same situation can be seen in the case of Figure 6, Figure 11, and Figure 12.
  7. An explanation is required why the results devoted to 50g of PP have been presented in Figure 2, while in Figure 3 information related only to 20g of the sample has been included?
  8. The DSC thermograms of PP after conditioning at various parameters should be added as supplementary materials.
  9. In the case of section 3.1.4 (line 280), there is no discussion of the obtained results. I suggest removing this part of the manuscript.
  10. (Line 300) Authors claim that the “Expandable graphite is a very effective and versatile flame retardant.” The appropriated references to sources describing expandable graphite as flame retardant have to be indicated.
  11. Figure 19 is barely legible because of a large number of results, with the names of samples being almost impossible do decipher. I suggest splitting Figure 19 into a few additional figures with regard to the type of additive (mineral additives, or possibly with and without an additive).
  12. Using the “MMT” designation in the manuscript is misleading because only a modified variant of MMT was used, not the pure one.
  13. The abbreviations PP, PP1, and PP2 have been used inconsistently and as a result, it is impossible to determine whether they refer to the same polypropylene variant, or if different types of PP were applied?
  14. In the manuscript, there was described that PP was crosslinked with silane und peroxide (lines 294-295). It has to be clearly indicated which of them was used during the formation if samples presented in Table 5. If the mixture of crosslinked agents was applied the composition of the mixture has to indicate.
  15. It is not possible to determine the meaning of the FR1, FR2, FR3, FR4, FR5, FR6 Ad1, PP1, PP2, Syn and other abbreviations used in table 5. The Authors claim that “The flame retardants, mentioned in Table 5, are different phosphorous based flame retardants.” This information, however, is insufficient from a scientific point of view. The composition of samples should be described in a way that allows repeating the performed analyses in subsequent studies.
  16. In Figure 21 the name of samples presented in table 5 should be used, not the percentage composition – it is difficult to recognize the specific samples. Moreover, the size of the font used is too small.
  17. In the “Conclusion” section of the paper, the Authors indicated that “The mass in both cases was slightly different and therefore the results cannot be directly compared” (lines- 450-451). If the results are not comparable there is no point in presenting and discussing them.
  18. (Lines 457-458) The Authors claim that “The heat release rates of PE-LD from different manufacturers showed slightly different behawior” There is no explanation as to what “different behavior” actually means in this context.
  19. In the submitted manuscript a huge number of samples and results have been presented. The manuscript should be divided into at least two papers. Otherwise, problems with the descriptions of the analyzed samples and the performed analyses will continue to arise. Moreover, the obtained results should be correlated with the XRD or TEM analysis in an aim to establish the structure of the obtained materials, which is crucial.
  20. Taking into account the inaccuracies mentioned above, unfortunately, I cannot recommend the submitted manuscript for publication.

Author Response

Comments to reviewer 2

  1. The designations of the analyzed polymers have to be indicated in the title of the manuscript. Otherwise, the reader can be misled into thinking that all types of polymers have been studied.

We added the information in the title.

  1. The introduction provides no viable information on the materials used in the study. A review of literature, devoted to the properties of flame retardants used in the research has to be included.

The materials are now described and extensive cross references were added.

  1. More information related to the mineral additives applied in the research is required. For example, the Authors reported having used MMT – Nanofil5. It is well known that Nanofil 5 is a modified variant of MMT. Taking into account that a modifier significantly influences the structure and properties of materials, modified and unmodified samples have to be analyzed and compared. Moreover, the name and the structure of the modifying agent has to be described.

As the influence of the formation of inorganic additives is not in the focus of this article. It shall only be shown what effect can be caused by such flame retardants. But we added the annotation that the structure may cause differences in the flame ability (line 386).

  1. Figure 3 – the same chart is presented twice. The same situation can be seen in the case of Figure 6, Figure 11, and Figure 12.

We are not sure about this comment, we believe this is a result of the word correction mode.

Figure 3 compares the HRR after one and 33 days after production of PP.

Figure 6 compares the HRR after on and 99 days after production of PP.

We do not think it will help if all curves are displayed in one diagram.

Figure 11 shows the HRR of the different PE-LD materials. If you mean the small decrease of the green curve at about 100 seconds, this is not at all comparable to the curve characteristic before. While the HRR of PP rises about 50 kW/m² to 100 kW/m², PE-LD shows only a small difference in the range of less than 10 kW/m²:

This is also valid for the blue curve in Figure 12.

  1. An explanation is required why the results devoted to 50g of PP have been presented in Figure 2, while in Figure 3 information related only to 20g of the sample has been included?

This is due to the fact that Figure 2 shows the HRR for the pressed samples and Figure 3 shows the HRR for the injection moulded samples. This is already described in the text.

  1. In the case of section 3.1.4 (line 280), there is no discussion of the obtained results. I suggest removing this part of the manuscript.

Although we have no detailed information about the polymers, we think it is important to show that polymers of the same type can show a slightly different burning behaviour. Though, we tried to improve this point.

  1. (Line 300) Authors claim that the “Expandable graphite is a very effective and versatile flame retardant.” The appropriated references to sources describing expandable graphite as flame retardant have to be indicated.

The references are now given in chapter 1.

  1. Figure 19 is barely legible because of a large number of results, with the names of samples being almost impossible do decipher. I suggest splitting Figure 19 into a few additional figures with regard to the type of additive (mineral additives, or possibly with and without an additive).

We have improved Figure 18 and Fig 19

  1. The abbreviations PP, PP1, and PP2 have been used inconsistently and as a result, it is impossible to determine whether they refer to the same polypropylene variant, or if different types of PP were applied?

PP1 and PP2 only indicate different PP types.

  1. In the “Conclusion” section of the paper, the Authors indicated that “The mass in both cases was slightly different and therefore the results cannot be directly compared” (lines- 450-451). If the results are not comparable there is no point in presenting and discussing them.

That is right but we explained that the behaviour is similar, although they have different masses. We did not compare them anyway but only pointed out that their burning behaviour is alike.

  1. (Lines 457-458) The Authors claim that “The heat release rates of PE-LD from different manufacturers showed slightly different behawior” There is no explanation as to what “different behavior” actually means in this context.

We clarified this by changing the sentence (“…slightly different curve characteristic.”).

  1. In the submitted manuscript a huge number of samples and results have been presented. The manuscript should be divided into at least two papers. Otherwise, problems with the descriptions of the analyzed samples and the performed analyses will continue to arise. Moreover, the obtained results should be correlated with the XRD or TEM analysis in an aim to establish the structure of the obtained materials,

which is crucial.

We tried our best to give a good overview about some polymers and various additives that can be used as flame retardants. Since it is not our intention to analyse the structure of the additives or compounds but to indicate possible effects in general, we think XRD or TEM would exceed this intention.

  1. Taking into account the inaccuracies mentioned above, unfortunately, I cannot recommend the submitted manuscript for publication.

We are thankful for your feedback and the improvement of our paper that was induced by it. Unfortunately, not all questions could be answered by this paper.

Reviewer 3 Report

Can be accept now.

Author Response

Thanks

Round 3

Reviewer 2 Report

1. The review of the literature, devoted to the properties of flame retardants used in the research is still poor. There is still lack of information why such flame retardants have been taken into account.

2. In the Respond Authors claim that Fig 19 has been improved. I do not see any changes. Figure 19 is still barely legible because of a large number of results, with the names of samples being almost impossible to decipher. I still suggest splitting Figure 19 into a few additional figures with regard to the type of additive (mineral additives, or possibly with and without an additive).

Taking into account the inaccuracies mentioned above I suggest a minor revision of the submitted manuscript

Author Response

Please see the reply for the reviewer in the attached document.
